# The epidemiology of gestational diabetes, gestation hypertension, and anemia in North Palestine from 2018 to 2020: A retrospective study

**Basma Damiri**[1], **Haytham Abumohsen**[2]* , **Souad BelKebir**[3], **Mahmoud Hamdan**[4], **Lubna Saudi**[3], **Hiba Hindi**[2], **Rawnaq Abdaldaem**[2], **Baraa Bustami**[2], **Abeer Almusleh**[2], **Osama Yasin**[2]

1 Medicine & Health Science Faculty, Drug, and Toxicology Division, An-Najah National University, Nablus, Palestine, 2 Medicine & Health Science Faculty, Department of Medicine, An-Najah National University, Nablus, Palestine, 3 Medicine & Health Science Faculty, Family and Community Medicine, An-Najah National University, Nablus, Palestine, 4 Medicine & Health Science Faculty, Graduate School, Clinical Laboratory Science Program, An-Najah National University, Nablus, Palestine

☯ These authors contributed equally to this work.
* hithammohamad97@gmail.com

**Data Availability Statement:** Most data generated or analyzed during this study are included in this manuscript. Other data supporting this study's

## Abstract

### Background

In Palestine, women face a challenging environment and a demanding lifestyle, which puts them at a higher risk of experiencing complications during pregnancy. This study aimed to examine the risk factors linked with abnormal hemoglobin (Hb) levels, gestational diabetes (GD), and gestational hypertension (GH) among pregnant women. The results was compared between women residing in cities or villages areas and those in refugee camps.

### Methods

Medical records (N = 7889) for pregnant women at primary healthcare centers in the North West Bank were reviewed for sociodemographic and medical data along with the reported fasting blood sugar, hemoglobin, and blood pressure in the first and second trimesters from July 2018 to July 2020. However, only 6640 were included in the analysis as 1249 were excluded for having multiple pregnancy or lost to follow up. Complications and risk factors were defined according to the available global guidelines. Then, descriptive analysis was used to show the percentages of different risk factors and complications among them. The correlation between the several characteristics and variables with these complications was assessed by calculating the odds ratios (OR) using logistic regression. P-values of <0.05 were considered significant.

### Results

The prevalence of adolescent pregnancy was the highest among women living in rural areas (9.8%) and grand multigravidity among refugee women (22%). The overall prevalence

findings and /or analyzed during the current study are available from the corresponding author upon reasonable request.

**Funding:** The author(s) received no specific funding for this work.

**Competing interests:** The authors have declared that no competing interests exist.

of anemia was higher in the second-trimester (16.2%) than in the first-trimester (11.2%), with anemic women in the first-trimester being more likely to be anemic in the second-trimester (OR = 8.223, P-value<0.001). Although anemia was less common in the first-trimester among refugees than among women living in urban areas (OR = 0.768, P-value = 0.006), it was more prevalent in the second-trimester (OR = 1.352, P-value<0.001). Moreover, refugee women were at lower risk than women living in urban areas of having GD (OR = 0.687, P-value<0.001) and diabetes mellitus (OR = 0.472, P-value<0.001) in the second-trimester. GH was associated with GD (OR = 1.401, P-value = 0.003) and DM (OR = 1.966, P-value<0.001).

## Conclusion

The findings of this study provide evidence-based data on the considerable prevalence of pregnancy complications, anemia, GD, and GH among Palestinian pregnant women living in the north of the West Bank. Multi gravida, gestational age, Hb levels, and the type of settings were strong predictors of pregnancy complications. Therefore, a national plan is needed to ensure adequate maternal care for all, especially disadvantaged women, those in rural areas and refugee camps.

## Introduction

Pregnancy complications contribute significantly to maternal, fetal, and neonatal morbidity and mortality [1]. Gestational diabetes (GD), gestational hypertension (GH), and abnormal hemoglobin (Hb) levels are the most common complications during pregnancy. Research has indicated a significant rise in the occurrence of GD and GH over the last twenty years. Additionally, these conditions are affecting women at a younger age, and the associated complications are becoming more severe [2, 3]. GD is currently the most common obstetrical complication, affecting about 14% of pregnant women worldwide, according to the International Diabetes Federation (IDF) [2, 3]. Moreover, GH accounts for 10% of hypertensive disorders during pregnancy [4]. It complicates 6–8% of pregnancies and causes significant maternal and fetal morbidity and mortality [5].

Studies showed that anemia and high hemoglobin (Hb) levels during pregnancy affect the quality of the mother's delivery and her baby's health. Pregnancy-induced anemia could lead to several gestational complications, including preterm birth, hypertensive disorders of pregnancy, and low birth weight [6–8]. On the other hand, previous studies demonstrated that elevated Hb levels in the first trimester indicate possible pregnancy complications and should not only be considered as good iron status [9, 10]. Therefore, early detection of Hb levels and treatment of pregnancy anemia are essential as they could lead to noticeable improvements in pregnancy outcomes [11].

The World health organization (WHO) reported that most pregnancy-related maternal deaths occur in low-resource settings and developing countries, where they could have been prevented [12]. As a result of their stressful living conditions, Palestinian women, especially those living in rural areas and refugee camps, maybe at a heightened risk of pregnancy complications [13, 14]. Palestinian refugees have been displaced internally and resided in refugee camps in the Gaza Strip and West Bank since the Israeli-Palestinian conflict began in 1948. These camps are managed by the United Nations Relief and Works Agency for Palestine

Refugees in the Near East (UNRWA) [13, 14]. According to the Palestinian Ministry of Health (MOH), the maternal mortality rate has increased up to four folds from 2017 (5.9 per 100,000 live births) to 2020 (28.5 per 100,000 live births) [15]. This high maternal morbidity and mortality burden for Palestinian women suggests further investigation [16]. Early diagnosis of pregnancy complications can improve prenatal care for pregnant women and result in a satisfactory pregnancy outcome [17, 18]. In addition, monitoring complications related to pregnancy in the West Bank on a regular, reliable, and routine basis can help identify problems and develop evidence-based interventions to stop, contain, and address existing and potential future problems.

Studies investigating Palestinian women's maternal health status and risk factors associated with adverse pregnancy outcomes are scarce which highlighted the great need to conduct such kind of research. These studies and reports have stated that Palestinian population has high prevalence of several pregnancy complications including gestational diabetes and hypertension but have never stated the exact percentages or addressed the living and socioeconomic circumstances as risk factors [15, 16, 19]. In addition, there is an obvious lack of research on the occurrence of various hypertensive disorders of pregnancy (HDP) in Palestinian women; they are not even included in the Palestinian health annual reports published by the Palestinian ministry of health (MOH) [20]. Therefore, this study aimed to investigate risk factors associated with main pregnancy complications, including abnormal Hb levels, GD, and GH among registered pregnant women at primary healthcare centers in the north West Bank from the July, 2018 to the July 2020. It also investigated and compared the health status of pregnant women regarding abnormal gestational Hb levels, fasting blood sugar (FBS), and blood pressure (BP) in the first-trimester (<13 gestational weeks (GW)) and the second-trimester (24GW). In addition, this study intended to compare results between the attendees based on their living type of setting; urban areas, rural areas, and refugee camps.

## Methodology

### Study design, setting, and population

This is a retrospective study in which we have collected data from medical records at Ministry of Health primary healthcare centers (MOH PHC) in the north of West Bank (Nablus, Jenin, Tubas, and Tulkarm) from the 1$^{st}$ of July 2018 to the 1$^{st}$ of July 2020. Data collection took approximately 1 year (accessed from 1/7/2019 until 1/7/2020 for research purposes only) The north West Bank represents around 40% of the total number of the West Bank residents. Three of the studied governorates are the largest governorates in the north, and the geographical distribution of these governorates covers around 85% the north West Bank. The MOH PHC clinics were chosen for this study as 70% of pregnant women in Palestine visit MOH PHC each year. Therefore, the researcher could easily access the pregnant women's data. A part of this study was during the COVID-19 pandemic and some pregnant women could not access healthcare services due to the partition and isolation of Palestinian villages. According to the MOH annual health reports for 2019 and 2020, the number of beneficiaries who received antenatal care at MOH clinics decreased by half during 2020 [21]. The total number of pregnant women registered (first visit) in the MOH PHC centers in the West Bank was 34605, 36605, and 28547 in 2018, 2019, and 2202, respectively [22]. The total reviewed medical reports were 7889 files. Any pregnant women with multiple pregnancies (N = 172) or those who had an abortion or lost to follow-up (N = 1077) were excluded from the study. Files for pregnant women with singleton pregnancies (N = 6640) were included and they represented around 15% of all cases registered in MOH PHC in the studied areas.

Regarding missing data, the main objective of our study was to determine the epidemiology of various pregnancy complications, which primarily depends on a single diagnostic value that is unrelated to other values. Therefore, all the available data in the files were included to generalize the results as much as possible. Additionally, we found that most pregnant women with missing data (N = 2565) have just one missing value, and excluding about a third of the sample due to a single missed value could negatively affect our sample representativeness of the study population. Some records lacked diagnostic values, such as FBS (n = 344), Hb (n = 266), or systolic or diastolic blood pressure at registration or second-trimester visits (n = 1438). In addition, some clinical and sociodemographic values such as gravity (n = 436) and age at gestation (n = 1) were also missing. Therefore, the diagnosis was made separately depending on the available values for each complication.

Pregnant women should follow up monthly with the antenatal clinics; however, some of them have attended their booking appointments only and never came back. We have tried to include every beneficial pregnant to the sample to make it as general as possible so we have included any pregnant women who at least attended the booking and a late trimesters appointments. Even though the number of excluded pregnant women who has lost to follow-up seems significant but it was not possible to avoid it. These cases had either only an initial visit or no third trimester visit which will not add any value to the results as pregnancy complication are diagnosed after 20 weeks of gestation.

## Operational definitions

Hemoglobin levels: Hb is indirectly used to diagnose anemia [23]. Based to the World Health Organization (WHO) and the U.S. Centres for Disease Control and Prevention (CDC) guidelines, anemia in pregnancy has different cut-offs based on the trimester (first trimester: <11.0 g/dl; second trimester: <10.5 gram per deciliter (g/dl); and third trimester: <11 g/dl) [24] while normal values assigned from 11-<12.5 g/dl [25]. Gestational diabetes: Pregnant women are diagnosed with GD if their FBS levels in the second-trimester are (92-<126 milligram per deciliter (mg/dl)) and will be considered to have diabetes mellitus (DM) if FBS $\geq$126 mg/dl [16]. Gestational hypertension: WHO defines GH as a condition in which the systolic blood pressure (SBP) $\geq$140 millimeter mercury (mmHg) or the diastolic blood pressure (DBP) $\geq$90 mmHg on two separate measurements, taken at least four to six hours apart after the 20th week of pregnancy [5]. Maternal and gestational ages: Maternal age in years was the age of the pregnant woman at her registration visit in the first-trimester ($\leq$13 weeks). To make it more clear the way of variables measurement was presented in a tabular formation in figure [1]. Advanced maternal age was defined as maternal age $\geq$35 years at registration [26], while adolescent pregnancy was defined as maternal age<20 years at registration [27], and age between 20 and 34 was considered childbearing age [27]. Gestational age in weeks was based on ultrasound estimation in the first visit in the first-trimester. Gravidity: Gravidity was categorized into primigravida (first pregnancy), multigravida (has been pregnant two to four times), and grand multigravida (has been pregnant five times or more) [28]. Types of living settings: The settings, or areas, were divided into three areas; urban, rural, and refugee camp (Fig 1).

## Data analysis

We used descriptive statistics to characterize the sample. Valid percentages were reported. Chi-squared and Fisher exact tests were used to estimate the statistically significant difference between categorical variables, and t-test and ANOVA were used to assess the difference between continuous variables. Tukey's test was applied for post hoc analysis. Multinomial logistic regression analysis models were conducted to determine the association between

| | | | |
|---|---|---|---|
| | FBS ≥126 | mg/dl | Diabetes Miletus (DM) |
| Fasting Blood Sugar (FBS) | FBS 92–<126 | mg/dl | Gestational Diabetes (GD) |
| | FBS <92 | mg/dl | Normal |
| | Hb <11 | g/dl | Anemia |
| Hemoglobin (Hb) during first and third trimester | Hb 11–<12.5 | g/dl | Normal |
| | Hb ≥12.5 | g/dl | High |
| | Hb <10.5 | g/dl | Anemia |
| Hemoglobin (Hb) during second trimester | Hb 10.5–<12.5 | g/dl | Normal |
| | Hb ≥12.5 | g/dl | High |
| Systolic Blood Pressure (SBP) | SBP ≥140 | mmHg | Elevated |
| | SBP <140 | mmHg | Normal |
| Diastolic blood pressure (DBP) | DBP ≥90 | mmHg | Elevated |
| | DBP <90 | mmHg | Normal |
| Abbreviations: mg/dl: milligram per deciliter, g/dl: gram per deciliter, mmHg: millimeter of mercury | | | |

**Fig 1. Operational definitions of the study variables according to the used global guidelines.**

maternal Hb levels in the first-trimester (Molde1), Hb levels in the second-trimester (Model 2), FBS in the second-trimester (Model 3), and the associated independent variables. Odds ratios (ORs) and their 95% confidence intervals (CIs) were used as indicators of levels of association. A p-value <0.05 was considered statistically significant. The analyses of the results also proceeded with the use of the Statistics Package for Social Sciences (SPSS. Version 22), adopting a significance level of 5%.

## Ethical considerations

The study was carried out following the ethical standards, Declarations of Helsinki. Approval was obtained from Institutional Review Board "IRB" at An-Najah National University (ANNU) in Palestine prior to the research conduction (Archive No. 10, the 13th of May, 2019). Additional approval to access medical fields from the Palestinian Ministry of Health was obtained. All data were collected and treated confidentially, kept safe. Personal and medical information was collected and analyzed anonymously. Codes were used instead of names to ensure confidentiality. Authors did not have access to information that could identify individual participants during or after data collection.

## Results

### Maternal background information

This table describes the sociodemographic data of the study population. Out of the 6640 included files, 55.3% were women living in urban areas, 14.0% were women living in rural areas, and 30.7% were refugees. Adolescent pregnancy was the highest among women living in rural areas (9.8%), while advanced maternal age pregnancy was the highest among women living in urban areas (11.8%) (P-value<0.001). In addition, the percentage of grand multigravidity was the highest among refugee pregnant women (22%), and that of multigravidity was the highest among women living in urban areas (67.6%) (P-value<0.001) (Table 1).

**Table 1. General characteristics of pregnant women based on the location.**

| Location | | Total | Refugee camps | Urban | Rural | P-value |
|---|---|---|---|---|---|---|
| | | n(%) | n(%) | n(%) | n(%) | |
| Age of pregnant women | Adolescent pregnancy | 505(7.6) | 116(5.7) | 298(8.1) | 91(9.8) | <0.001 |
| | Childbearing age | 5378(81.0) | 1706(83.6) | 2937(80.1) | 735(79.0) | |
| | Advanced maternal age | 756(11.4) | 219(10.7) | 433(11.8) | 104(11.2) | |
| Gravidity | Primigravida | 1480(23.1) | 403(19.8) | 813(23.0) | 264(31.8) | <0.001 |
| | Multigravida | 4086(63.8) | 1189(58.3) | 2387(67.6) | 510(61.4) | |
| | Grand multigravida | 638(13.1) | 448(22.0) | 331(9.4) | 57(6.9) | |

Abbreviations: n: number, %: percentage.

## General maternal information in the first and second trimesters

The following table describes the general characteristics of pregnant women in the first and second trimesters. Refugee pregnant women have a higher mean of gravidity (3.25±2.1) compared to women living in urban areas (2.58±1.6) and women living in rural areas (2.22±1.4) (P-value <0.001). In addition, pregnant refugee women at registration had significantly higher mean values of FBS (91.39±14.8 mg/dl) and Hb levels (12.02±1.2g/dl) than women living in urban areas (88.46±14.8 mg/dl) (P-value<0.001). Refugee women also have lower SBP (11.54 ±12.0 mmHg) and DBP (67.13±9.2mmHg) compared to women living in urban areas (71.79 ±10.7 mmHg) and women living in urban areas (73.25±210.9 mmHg) (P-value≤0.014). In the second trimester, refugee pregnant women had significantly lower mean levels of FBS, Hb, SBP, and DBP compared to women living in urban and rural areas (P-value <0.001) (Table 2).

## Biomedical information

The third table describes the abnormal FBS, Hb, and blood pressure in the first and second-trimesters. In the first trimester, anemia was more prevalent among women living in urban areas (12.3%) than refugees (9.8%) and women living in rural areas (9.7%). Refugee women (36.2%) were more likely to have high Hb levels than women living in rural areas (35.4%) and women

**Table 2. General characteristics of pregnant women in the first and the second-trimesters.**

| Time | | Mean (SD) | | | | P-value | P-value | P-value |
|---|---|---|---|---|---|---|---|---|
| | | Total | Refugee camp | Urban | Rural | Camp vs Urban | Camp vs Rural | Urban vs Rural |
| At registration | Maternal age (years) | 26.88(5.8) | 27.18(5.5) | 26.79(5.9) | 26.57(6.1) | <0.001 | <0.001 | 0.99 |
| (<13GW) | Gestational age (weeks) | 9.25(5.5) | 10.55(9.0) | 8.68(2.7) | 8.65(2.5) | <0.001 | <0.001 | 0.742 |
| | Gravidity | 2.75(1.8) | 3.25(2.1) | 2.58(1.6) | 2.22(1.4) | <0.001 | <0.001 | <0.001 |
| | FBS (mg/dl) | 89.72(15.0) | 91.39(14.8) | 88.46(14.8) | 91.03(16.2) | <0.001 | 0.823 | <0.001 |
| | Hb (g/dl) | 11.94(1.2) | 12.02(1.2) | 11.89(1.2) | 11.97(1.2) | <0.001 | 0.518 | 0.136 |
| | SBP (mmHg) | 110.82(11.6) | 109.52(10.4) | 111.54(12.0) | 110.82(12.5) | <0.001 | 0.014 | 0.212 |
| | DBP (mmHg) | 70.65(10.5) | 67.13(9.2) | 71.79(10.7) | 73.25(10.9) | <0.001 | <0.001 | <0.001 |
| At 24 GW | FBS (mg/dl) | 92.82(19) | 90.72(16.0) | 93.25(20.0) | 95.42(20.3) | <0.001 | <0.001 | 0.006 |
| | Hb (g/dl) | 11.74(1.3) | 11.60(2.9) | 11.86(1.3) | 11.75(1.2) | <0.001 | <0.001 | 0.03 |
| | SBP (mmHg) | 114.35(16.8) | 112.19(22.5) | 115.43(13.3) | 114.75(13.9) | <0.001 | <0.001 | 0.327 |
| | DBP (mmHg) | 72.75(11.6) | 67.86(11.0) | 74.71(11.0) | 75.55(11.7) | <0.001 | <0.001 | <0.001 |

Abbreviations: Hb: hemoglobin, GW: gestational week, FBS: fasting blood sugar, SBP: systolic blood pressure, DBP: diastolic blood pressure, mg/dl: milligram per deciliter, g/dl: gram per deciliter, mmHg: millimeter of mercury, SD: standard deviation, n: number.

**Table 3. The prevalence of FBS, hemoglobin, and blood pressure categories in the first and the second-trimesters.**

| | Variables | Category | Total | Refugee camp | Urban | Rural | P value |
|---|---|---|---|---|---|---|---|
| | | | n(%) | n(%) | n(%) | n(%) | |
| At registration (<13GW) | FBS (mg/dl) | ≥ 126 | 74(1.1) | 23(1.1) | 29(0.8) | 22(2.4) | <0.001 |
| | | 92–125 | 2602(39.6) | 877(43.1) | 1330(36.8) | 395(42.9) | |
| | | <92 | 3897(59.3) | 1135(55.8) | 2259(62.4) | 503(54.7) | |
| | Hb (g/dl) | <11 | 735(11.2) | 200(9.8) | 446(12.3) | 89(9.7) | 0.004 |
| | | 11–12.49 | 3596(54.8) | 1096(53.9) | 1997(55.2) | 503(54.9) | |
| | | ≥12.5 | 2237(34.1) | 736(36.2) | 1177(32.5) | 324(35.4) | |
| | SBP (mmHg) | ≥140 | 67(1.0) | 16(0.8) | 43(1.2) | 8(0.9) | 0.332 |
| | DBP (mmHg) | ≥90 | 247(3.8) | 30(1.5) | 176(4.8) | 41(4.5) | <0.001 |
| At 24 GW | FBS (mg/dl) | ≥ 126 | 393(6.2) | 78 (4.2) | 246(6.8) | 69(7.6) | <0.001 |
| | | 92–125 | 2455(38.6) | 597(32.5) | 1445(39.9) | 413(45.6) | |
| | | <92 | 3515(55.2) | 1164(63.3) | 1927(53.3) | 424(46.8) | |
| | Hb (g/dl) | <10.5 | 1045(16.2) | 372(19.5) | 528(14.5) | 145(16.1) | <0.001 |
| | | 11–12.49 | 3509(54.4) | 1127(59.0) | 1890(52.0) | 492(54.5) | |
| | | ≥12.5 | 1892(29.4) | 412(21.6) | 1215(33.4) | 265(29.4) | |
| | SBP (mmHg) | ≥140 | 201(3.1) | 60(3.0) | 106(2.9) | 35(3.9) | 0.345 |
| | DBP (mmHg) | ≥90 | 449(6.9) | 71(3.6) | 287(7.9) | 91(10.0) | <0.001 |

Abbreviations: Hb: hemoglobin, FBS: fasting blood sugar, GW: gestational week, SBP: systolic blood pressure, DBP: diastolic blood pressure, mg/dl: milligram per deciliter, g/dl: gram per deciliter, mmHg: millimeter of mercury, n: number.

living in urban areas (32.5%) (P-value = 0.004). More women living in urban areas (4.8%) and women living in rural areas (4.5%) had higher DBP than refugees (1.5%). (P-value <0.001). The prevalence of DM was higher among women living in rural areas in the first and the second-trimesters (2.4%, 7.6%), respectively than among refugee women (1.1%, 4.2%) and women living in urban areas (0.8%, 6.8%) (P-value <0.001).

In the second trimester, GD was more prevalent among women living in rural areas (45.6%) than urban (39.9%) and refugee women (32.5%) (P-value <0.001). Pregnant women living in refugee settings had a higher prevalence of anemia (19.5%) than women living in rural (16.1%) and urban settings (14.5%) (P-value<0.001). Pregnant women living in rural settings were more likely to have high DBP (10%) than in urban (7.9%) and refugee settings (3.6%) (P-value<0.001) (Table 3).

### Adjusted logistic regression models

**Model 1.** The results of the adjusted multinomial logistic regression analysis for the risk factors of abnormal maternal Hb levels (low and high Hb) in the first the first-trimester revealed that pregnant women living in refugee camps were at lower risk of being anemic (AOR = 0.768, P-value = 0.006) and at higher risk of having increased Hb levels (AOR = 1.190, P-value = 0.006) than women living in urban areas. In addition, primigravida (AOR = 0.567, P-value<0.001) and multigravida (AOR = 0.673, P-value = 0.001) were associated with lower risk of anemia, while gestational age was associated with decreased risks of high Hb levels (AOR = 0.980, P-value = 0.009) (Table 4).

**Model 2.** The results of the adjusted multinomial regression model for the risk factors associated with abnormal Hb levels (low and high) in the second-trimester revealed that refugee pregnant women were more likely to be anemic (AOR = 1.352, P-value<0.001) and less likely to have high Hb levels (AOR = 0.573, P-value<0.001) than women living in urban areas.

**Table 4. Multinomial regression model for predictors associated with abnormal hemoglobin levels (low and high) in first-trimester.**

| Hb status | Covariate | Covariate categories | AOR | 95% CI | P-value |
|---|---|---|---|---|---|
| Low Hb (<11 g/dl)* | Location | Camp | 0.768 | 0.635–0.929 | 0.006 |
| | | Rural | 0.826 | 0.635–1.075 | 0.155 |
| | | Urban (reference) | | | |
| | Age | Adolescent pregnancy | 0.975 | 0.640–1.486 | 0.908 |
| | | Childbearing age | 1.022 | 0.786–1.330 | 0.868 |
| | | Advanced maternal age (reference) | | | |
| | Gravidity | Primigravida | 0 .567 | 0.420–0.766 | <0.001 |
| | | Multigravida | 0.673 | 0.528–0.857 | 0.001 |
| | | Grand multigravida (reference) | | | |
| | Gestational age | | 1.008 | 0 .997–1.018 | 0.143 |
| High Hb (≥12.5 g/dl) * | Location | Camp | 1.190 | 1.052–1.346 | 0.006 |
| | | Rural | 1.135 | 0.963–1.339 | 0.132 |
| | | Urban(reference) | | | |
| | Age | Adolescent pregnancy | 1.028 | 0.782–1.351 | 0.845 |
| | | Childbearing age | 1.030 | 0.857–1.239 | 0.749 |
| | | Advanced maternal age (reference) | | | |
| | Gravidity | Primigravida | 1.057 | 0.858–1.301 | 0.602 |
| | | Multigravida | 1.062 | 0.888–1.269 | 0.511 |
| | | Grand multigravida(reference) | | | |
| | Gestational age | | 0.980 | 0.965–0.995 | 0.009 |

*The reference category is normal Hb: 11–12.49 (g/dl).

Abbreviations: Hb: hemoglobin, FBS: fasting blood sugar, DBP: diastolic blood pressure, SBP: systolic blood pressure, mg/dl: milligram per deciliter, g/dl: gram per deciliter, mmHg: millimeter of mercury, AOR: adjusted odds ratio, CI: confidence interval.

Pregnant women with low Hb levels in the first-trimester were more likely to be anemic in the second-trimester (AOR = 8.223, P-value<0.001). Pregnant women in their childbearing age were less likely to have high Hb (AOR = 0.769, P-value = 0.009), while multigravida women were more likely to have a high Hb level in the second-trimester (AOR = 1.245, P-value = 0.034) (Table 5).

**Model 3.** The results of the adjusted multinomial logistic regression model for the risk factors associated with gestational diabetes in the second-trimester revealed that refugee women were at lower risk of GD (AOR = 0.687, P-value<0.001) and DM (AOR = 0.472, P-value<0.001) than women living in urban areas. High Hb levels in the first-trimester (≥12.5 g/dl) were associated with an increased risk of GD and DM (P-value ≤0.001). High DBP in the second-trimester was associated with an increased risk of GD (AOR = 1.401, P-value = 0.003). Multigravida was associated with an increased risk of GD (AOR = 1.215, P-value = 0.036) and DM (AOR = 1.498 P-value = 0.040). Adolescent pregnancy age (AOR = 0.573, P-value = 0.049) and childbearing age pregnancy (AOR = 0.623, P-value = 0.005) were associated with decreased risk of DM (Table 6).

## Discussion

In Palestine, women face a challenging environment and a demanding lifestyle, which puts them at a higher risk of experiencing complications during pregnancy [14, 29]. It is estimated that one out of every four pregnant women is classified as a high-risk pregnancy, which amounts to roughly 14,000 cases per year. As a result, these women are particularly susceptible

**Table 5. Adjusted multinominal logistic regression model for predictors associated with abnormal hemoglobin levels in second-trimester.**

| Hb status | Covariate | Covariate categories | AOR | 95% CI | P-value |
|---|---|---|---|---|---|
| Low Hb (<10.5 g/dl) * | Location | Camp | 1.352 | 1.149–1.590 | <0.001 |
| | | Rural | 1.115 | 0.887–1.402 | 0.351 |
| | | Urban(reference) | | | |
| | Hb at first-trimester(g/dl) | <11 | 8.223 | 6.472–10.449 | <0.001 |
| | | ≥12.5 | 1.941 | 1.596–2.361 | <0.001 |
| | | 11–12.49 (reference) | | | |
| | Age | Adolescent pregnancy | 1.101 | 0.764–1.588 | 0.606 |
| | | Childbearing age | 0.900 | 0.703–1.150 | 0.399 |
| | | Advanced maternal age (reference) | | | |
| | Gravidity | Primigravida | 0.961 | 0.732–1.264 | 0.778 |
| | | Multigravida | 1.019 | 0.811–1.281 | 0.869 |
| | | Grand multigravida (reference) | | | |
| High Hb (≥12.5 g/dl) * | Location | Camp | 0.573 | 0.498–0.660 | <0.001 |
| | | Rural | 0.809 | 0.675–0.969 | 0.022 |
| | | Urban(reference) | | | |
| | Hb at first-trimester(g/dl) | <11 | 0.388 | 0.302–0.498 | <0.001 |
| | | ≥12.5 | 0.391 | 0.346–0.442 | <0.001 |
| | | 11–12.49(reference) | | | |
| | Age | Adolescent pregnancy | 0.837 | 0.622–1.126 | 0.240 |
| | | Childbearing age | 0.769 | 0.630–0.938 | 0.009 |
| | | Advanced maternal age (reference) | | | |
| | Gravidity | Primigravida | 1.248 | 0.986–1.579 | 0.066 |
| | | Multigravida | 1.245 | 1.016–1.525 | 0.034 |
| | | Grand multigravida (reference) | | | |

Abbreviations: Hb: hemoglobin, FBS: fasting blood sugar, DBP: diastolic blood pressure, SBP: systolic blood pressure, mg/dl: milligram per deciliter, g/dl: gram per deciliter, mmHg: millimeter of mercury, AOR: adjusted odds ratio, CI: confidence interval.

to maternal mortality and require specialized healthcare during their pregnancy [30]. This study estimated the prevalence of common pregnancy complications, including GD, DM, GH, and gestational anemia. The results of this study have several clinical implications for early screening of pregnant women for pregnancy complications.

There were several remarkable findings in this study. First, the results revealed that the prevalence of GD and GH among Palestinian women was higher than in other Arab countries [31–33] and European ones [34, 35]. Therefore, a significant stride in addressing gestational diabetes and hypertension is needed. Advanced maternal age is significantly associated with adverse perinatal and obstetrical outcomes [36]. It has been reported through a recent study that the risk of developing DM increases significantly with maternal age, particularly in pregnant women who are over 35 years old [37]. Our results also indicated that pregnant women in adolescence and childbearing age have a lower risk for developing DM than pregnant in advanced maternal age (≥35 years at registration). Furthermore, in line with previous research, expecting mothers who experienced high diastolic blood pressure during their pregnancy are more likely to develop gestational diabetes [38, 39]. Until recently, healthcare providers in Palestinian MOH clinics did not have written protocols or guidance for managing or screening GH and GD [40]. This makes HDP one of the most important and unresolved issues in obstetrics. Additionally, Palestinian women face significant complications due to under-treatment of HDP [41]. The lack of prior research has made it difficult to compare these

**Table 6. Adjusted multinomial logistic regression model for predictors of gestational diabetes and diabetes mellitus in the second-trimester.**

| Diabetes Miletus | Covariate | Covariate categories | AOR | 95% CI | P-value |
|---|---|---|---|---|---|
| FBS (≥126 mg/dl)* | Location | Camp | 0.472 | 0.357–0.624 | <0.001 |
| | | Rural | 1.055 | 0.775–1.435 | 0.735 |
| | | Urban (reference) | | | |
| | Hb at first-trimester trimester (g/dl) | <11 | 0.395 | 0.261–0.597 | <0.001 |
| | | 11–12.49 | 0.319 | 0.252–0.403 | <0.001 |
| | | ≥12.5 (reference) | | | |
| | Age | Adolescent pregnancy | 0.573 | 0.329–0.998 | 0.049 |
| | | Childbearing age | 0.623 | 0.447–0.869 | 0.005 |
| | | Advanced maternal age (reference) | | | |
| | Gravidity | Primigravida | 1.114 | 0.701–1.770 | 0.649 |
| | | Multigravida | 1.498 | 1.019–2.202 | 0.040 |
| | | Grand multigravida (reference) | | | |
| | FBS in the first-trimester (mg/dl) | ≥126 | 14.727 | 7.017–30.906 | <0.001 |
| | | 92–125 | 3.790 | 3.014–4.765 | <0.001 |
| | | <92 (reference) | | | |
| | SBP in the second-trimester (mmHg) | ≥140 | 1.377 | 0.794–2.386 | 0.254 |
| | | <140 (reference) | | | |
| | DBP in the second-trimester (mmHg) | ≥90 | 1.966 | 1.356–2.850 | <0.001 |
| | | <90 (reference) | | | |
| Gestational Diabetes | Location | Camp | 0.687 | 0.605–0.780 | <0.001 |
| | | Rural | 1.057 | 0.895–1.249 | 0.514 |
| FBS (92–125 mg/dl) | | Urban (reference) | | | |
| | Hb at first-trimester (g/dl) | <11 | 0.982 | 0.814–1.185 | 0.850 |
| | | 11–12.49 | 0.815 | 0.723–0.919 | 0.001 |
| | | ≥12.5 (reference) | | | |
| | Age | Adolescent pregnancy | 0.855 | 0.648–1.127 | 0.266 |
| | | Childbearing age | 0.844 | 0.700–1.019 | 0.077 |
| | | Advanced maternal age (reference) | | | |
| | Gravidity | Primigravida | 1.235 | 0.999–1.527 | 0.051 |
| | | Multigravida | 1.215 | 1.013–1.458 | 0.036 |
| | | Grand multigravida (reference) | | | |
| | FBS in the first-trimester (mg/dl) | ≥126 | 3.256 | 1.767–5.999 | <0.001 |
| | | 92–125 | 2.157 | 1.929–2.413 | <0.001 |
| | | <92 (reference) | | | |
| | SBP in the second-trimester (mmHg) | ≥140 | 1.300 | 0.947–1.786 | 0.105 |
| | | <140 (reference) | | | |
| | DBP in the second-trimester (mmHg) | ≥90 | 1.401 | 1.119–1.753 | 0.003 |
| | | <90 (reference) | | | |

The reference value is FBS levels <92 mg/dl. Abbreviations: Hb: Hemoglobin, FBS: fasting blood sugar, DM: diabetes miletus, DBP: diastolic blood pressure, SBP: systolic blood pressure, mg/dl: milligram per deciliter, g/dl: gram per deciliter, mmHg: millimeter of mercury, first: first, second: second, AOR = adjusted odds ratio, CI: confidence interval.

findings with others. Overall, the relatively high prevalence of both GD and GH in this study requires a high standard level of care. This also raises the question of whether the healthcare provided was sufficient and of good quality [40, 42]. It accentuates the urgent need for revision of DM and GH screening protocols and a close clinical and biomedical follow-up for pregnant

women with such complications [16]. These results also revalue the need for further research regarding the cause of this high prevalence and to give more attention to their early management and follow-up.

Anemia is one of the common obstetrical and perinatal problems. It is associated with many serious complications like low birth weight and preterm birth [23]. It was demonstrated that the prevalence rates of anemia among pregnant Palestinian women in 2007 were more than two times higher than those observed in Europe [43]. A previous study recommended further studies to examine the relative risk and causality of developing pregnancy-induced anemia among Palestinian women [43]. In this study, the prevalence of anemia was close to the prevalence of other Arabian countries [44]. According to the WHO classification, anemia is identified as a mild public health problem when population studies indicate a prevalence of 5.0–19% [45]. The overall prevalence of anemia among pregnant women in this study was within this range and higher in the second-trimester (16.2%) than the first-trimester (11.2%). In agreement with other studies, anemic women in the first-trimester were eight times more likely to also be anemic in the second-trimester [46, 47]. These results indicate that low levels of Hb in the first trimester may indicate the presence of anemia in later pregnancy. Therefore, early detection is crucial to ensure optimal care and minimize complications for both the mother and fetus. This information highlights the need for auditing the iron and folic acid supplementation protocols application and checking the quality of the dietary education programs as auditing has shown significant improvements in other health sections [48].

While this study explicitly examines physiological anemia that peaks in the second trimester, the findings from the first trimester are also significant. Physicians and health care providers give more attention to maternal anemia than high blood levels. In normal pregnancy, blood volume expands by an average of 50% compared with the non-pregnant state [49]. In agreement with these studies, anemic pregnant women in this study had a lower risk of developing GD and overt diabetes [50–53]. Since these complications interfere with each other, more attention and screening for the other complications are needed. The findings indicate that using Hb levels as a screening tool for identifying GD and HDP and predicting adverse outcomes for both mothers and infants could be a valuable approach, particularly in developing nations.

Research has shown that pregnant women who live in rural and suburban areas tend to experience more pregnancy-related complications. They face a higher risk of developing GD and overt diabetes compared to their counterparts residing in urban areas [54–58]. In agreement with these studies, GD was more prevalent among women living in rural than urban areas and refugee camps. While there is a considerable global disparity in anemia prevalence between urban and rural areas, most literature suggests that anemia is more common in rural areas than in urban areas [59, 60]. Although pregnant women living in rural areas and refugee camps were less likely to be anemic in the first-trimester than women living in urban areas, they were more likely to be anemic in the second-trimester. Moreover, pregnant women from rural areas tend to have higher adolescent pregnancy rates than those living in urban areas and refugee camps. A possible explanation for this discrepancy is the differences in clinical practice and the socioeconomic differences between these areas. Rural areas have a higher risk of having severe maternal morbidity and mortality due to multiple clinical and social factors, including workforce shortages, poverty, food security, violence, and trauma [61]. Like in many other countries, obstetrical and gynecological care facilities in the West Bank are less common in rural areas, and visit rates are lower than in urban areas [62]. The COVID-19 pandemic has worsened the circumstances for pregnant women, especially those residing in rural areas. Half of the study time coincided with the pandemic, further limiting access to healthcare for women attending public sector services, and to a lesser extent, for those who could afford

private sector services [21]. Many pregnant women living in rural areas have been unable to receive healthcare services due to the lockdowns, partition, and isolation of Palestinian villages, closure of antenatal care and high-risk pregnancy clinics, as well as limited transportation options [21]. On the other hand, women residing in urban areas and refugee camps experienced fewer restrictions in accessing healthcare services during the pandemic. The importance of this result lies in anticipating a high prevalence of such complications in these vulnerable groups, especially during a crisis. Moreover, rural pregnant women are less likely to take advantage of offered care despite its adequacy and ease of access, exacerbated by social, educational, and economic conditions [56]. Additionally, awareness regarding maternal health, family planning, and different socioeconomic factors plays a significant role in the low access to maternal health services in rural areas [63]. Rural pregnant women have less awareness regarding obstetrical danger signs and complications than women living in urban areas, making them more vulnerable to late diagnosis of these conditions [64]. Intensive monitoring and specialized care should be used for these groups including improving the health care services given along with spending more time educating pregnant women regarding family planning and contraception.

Grand multiparty is associated with several gestational complications, including diabetes mellitus, hypertension, antepartum hemorrhage, and mal-presentation [65]. This study showed that grand multigravidity was the highest among pregnant refugee women and was associated with anemia. Studies have shown that refugee women experience higher rates of increased gravidity and multigravidas compared to women who are not refugees. For instance, increased mean gravidity among women living in refugee camps was also observed among Syrian refugee women live in Turkey compared to Turkish women [66]. Moreover, multigravidas were also more common among Afghan refugees than non-refugees [67]. Therefore, various strategies are required to enhance awareness and acceptance of care among pregnant women resident in rural areas and refugee camps.

The study demonstrates a clear link between the living setting and the probability of pregnancy complications, especially in rural areas and refugee camps. It also provides insight into the prevalence of different pregnancy complications and risk factors in Palestine's various living settings, serving as a valuable point of reference for decision-makers to implement suitable measures and assess their efficacy in the future. It is important to mention that the study utilized an extensive and representative research sample from the northern region of the West Bank. However, it's worth conducting further studies after the pandemic crisis and in the southern region of the West Bank as the results may vary.

One potential limitation of the study is the lack of information on pregnant women's body mass index and other socioeconomic factors in the medical records. Due to this, our ability to fully understand the increased occurrences of GD and GH within the studied population may be restricted. Moreover, the studied medical records did not contain some crucial measures, such as socioeconomic circumstances. We were also unable to evaluate other potential complications, such as pre-eclampsia, due to the lack of documentation regarding urine analysis and oral-glucose-tolerance-test results. In addition, no fetal or perinatal complications were recorded. However, this study included a prominent number of pregnant Palestinian women, which is unprecedented in previous studies. It is also the first one to study Palestinian society from all available aspects, including laboratory, clinical, obstetrical, and medical data along with social and living conditions. Then, we have linked these characteristics to the complications that it carries. Therefore, It will be the core for further studies monitoring Palestinian women's health and as an alert to give more attention to some complications in specific groups of the population. Also, actions should be taken urgently to reduce the high prevalence of these complications or at least address it early. Finally, this study is also a realistic example of the possibility of using the WHO guidelines in the Palestinian community.

## Conclusion

The findings of this study provide evidence-based data on the considerable prevalence of pregnancy complications, anemia, GD, and GH among Palestinian pregnant women living in the north of the West Bank and suggesting a substandard quality of care especially during crisis. The findings suggest that a low Hb level at registration could be utilized in predicting the risk of developing anemia later in pregnancy. Early detection of anemia in pregnancy could lead to more intensive care and follow-up, thus decreasing the mother and fetus's complication rates. Moreover, the results demonstrate a clear link between the living setting and the probability of pregnancy complications. The second-trimester anemia was more prevalent among women living in rural areas and refugee camps than their counterparts residing in urban areas. In contrast, first-trimester anemia was more common in pregnant women living in urban areas. On the other hand, GD and GH were more common in rural pregnant women than in others. Pregnant women in rural areas tend to have higher rates of adolescent pregnancy and pregnancy complications than women living in urban areas. Therefore, to reduce maternal morbidity and mortality rates, a national plan is needed to ensure adequate maternal care for all, especially disadvantaged women, those in rural areas and refugee camps. In addition, the findings emphasize the importance of implementing nationwide preconception care services to enhance maternal health before pregnancy. Promoting awareness campaigns concerning the numerous complications and risk factors linked to pregnancy is crucial. Emphasis should be placed on maternal education and family planning methods. It is recommended to prioritize and adopt a focused approach to address prevalent pregnancy complications in particular regions. Moreover, we recommend conducting further research to investigate the underlying reasons for the high prevalence of these complications in each living setting.

## Supporting information

**S1 Data.**
(XLSX)

## Acknowledgments

The authors are very thankful to all who helped us with data collection and facilitated the work of this study.

## Author Contributions

**Conceptualization:** Basma Damiri, Haytham Abumohsen, Mahmoud Hamdan, Lubna Saudi, Baraa Bustami.

**Data curation:** Basma Damiri, Haytham Abumohsen, Rawnaq Abdaldaem.

**Formal analysis:** Basma Damiri, Haytham Abumohsen, Souad BelKebir, Lubna Saudi, Osama Yasin.

**Funding acquisition:** Basma Damiri, Haytham Abumohsen, Mahmoud Hamdan, Lubna Saudi, Baraa Bustami, Osama Yasin.

**Investigation:** Basma Damiri, Haytham Abumohsen, Lubna Saudi, Rawnaq Abdaldaem, Abeer Almusleh, Osama Yasin.

**Methodology:** Basma Damiri, Haytham Abumohsen, Souad BelKebir, Rawnaq Abdaldaem, Baraa Bustami.

**Project administration:** Basma Damiri, Haytham Abumohsen, Mahmoud Hamdan.

**Resources:** Basma Damiri, Haytham Abumohsen, Souad BelKebir, Hiba Hindi, Baraa Bustami.

**Software:** Basma Damiri, Haytham Abumohsen, Mahmoud Hamdan, Hiba Hindi.

**Supervision:** Basma Damiri, Haytham Abumohsen, Hiba Hindi.

**Validation:** Basma Damiri, Haytham Abumohsen.

**Visualization:** Basma Damiri, Haytham Abumohsen, Hiba Hindi.

**Writing – original draft:** Basma Damiri, Haytham Abumohsen.

**Writing – review & editing:** Basma Damiri, Haytham Abumohsen.

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
