## [Decision Letter · Decision Letter 0]

18 Oct 2023

PONE-D-23-29769The epidemiology of gestational diabetes, gestation hypertension, and anemia in North Palestine from 2018 to 2020: A retrospective study.PLOS ONE

Dear Dr. Abumohsen,

Thank you for submitting your manuscript to PLOS ONE. After careful consideration, we feel that it has merit but does not fully meet PLOS ONE’s publication criteria as it currently stands. Therefore, we invite you to submit a revised version of the manuscript that addresses the points raised during the review process. Additionally , address the following queries-There is a lack detailed methodological information in the abstract More background information of the dataset could be addedPresent the key variables and how they are measured in a tabular formatIn adjusted model, kindly mention the variables were adjustedAdd some strengths after the limitationsPlease submit your revised manuscript by Dec 02 2023 11:59PM. If you will need more time than this to complete your revisions, please reply to this message or contact the journal office at plosone@plos.org. Please include the following items when submitting your revised manuscript:A rebuttal letter that responds to each point raised by the academic editor and reviewer(s). You should upload this letter as a separate file labeled 'Response to Reviewers'.A marked-up copy of your manuscript that highlights changes made to the original version. You should upload this as a separate file labeled 'Revised Manuscript with Track Changes'.An unmarked version of your revised paper without tracked changes. You should upload this as a separate file labeled 'Manuscript'.

We look forward to receiving your revised manuscript.

Kind regards,

Russell Kabir, PhD

Academic Editor

PLOS ONE

Journal Requirements:

Reviewers' comments:

Reviewer's Responses to Questions

**Comments to the Author**

1. Is the manuscript technically sound, and do the data support the conclusions?

Reviewer #1: Partly

Reviewer #2: Yes

Reviewer #3: Yes

2. Has the statistical analysis been performed appropriately and rigorously? 

Reviewer #1: No

Reviewer #2: Yes

Reviewer #3: I Don't Know

3. Have the authors made all data underlying the findings in their manuscript fully available?

Reviewer #1: Yes

Reviewer #2: Yes

Reviewer #3: Yes

4. Is the manuscript presented in an intelligible fashion and written in standard English?

Reviewer #1: No

Reviewer #2: Yes

Reviewer #3: Yes

5. Review Comments to the Author

Reviewer #1: Epidemiology is not clearly defined.Risk factors are not properly stated.Grammatic errors (Some sentence in first paragraph is in future tense)and mistake in certain areas regarding the year of study. Mistakes in Table headings. Results need more specific explanation on associated factors and riskfactors.

Reviewer #2: Introduction

1.Well written

2.Please emphasize the novelty of the study and provide two or more related works in the context of Palestine.

Methods

1.Data exclusion criteria explained well.

2.Please provide a detailed explanation for why we did not consider different available methods for imputing missing values in electronic health data or health data records, given that missing data in your final samples is a significant concern.

3.Extend Data analysis a bit more, with more specification.

Results- Well written

Discussion

1.Well written

2.Please make it a bit short in length and focus on your objective and future implication in terms of Palestine.

Conclusion – well written

Reviewer #3: Abstract: Topic sentence(1st sentence) should rewrite for impressive/catchy

The results will be compared between women residing in urban or rural areas and those in refugee camps- Should change as the study has been already completed, however mentioning urban or rural -which should be clearly defined .

Introduction: dissimilar font size- please check carefully whole manuscript.

Methodology: The approach and procedure for gathering data for a retrospective medical study are explained in this section. The study has a few weaknesses in addition to its strengths as follows

1.Data Collection Period: From July 2018 to July 2020, research data were gathered. Although this is a significant amount of time, a longer duration of data collecting would have strengthened the results as a retrospective study This would give a more thorough grasp of the patterns and alterations across time.

2. The study was carried out during the pandemic covid -19, which had a major effect on patient accessibility and healthcare services. How this might have impacted the data and whether any modifications were taken to take these particular conditions into account are not addressed in detail in the study.

3. The study's exclusion criteria, which include removing expectant mothers who are carrying multiples, have undergone an abortion, or are lost to follow-up, may result in selection bias. The selection criteria for this exclusion may not have been well thought through, and it's probable that pregnant women in general are not adequately represented by it.

4.The study notes that there was a significant decrease in the number of beneficiaries receiving antenatal care in 2020. This decrease could be a crucial factor affecting the study's results, but the passage does not elaborate on how this change was addressed in the analysis.- Temporal changes

The limitations of the study should mentioned.

6. PLOS authors have the option to publish the peer review history of their article (what does this mean?). If published, this will include your full peer review and any attached files.

Reviewer #1: No

Reviewer #2: **Yes: **Md Karimuzzaman

Reviewer #3: No

---

## [Author Response · Author response to Decision Letter 0]

20 Dec 2023

Responds to reviewers and editors

Hello editor/s 

I am Haytham Abumohsen, the corresponding author for this research. First of all, thank for all of you editors and reviewers for your time and efforts evaluating and checking our manuscript, your effort and time are highly appreciated. Second i tried my best to do the requested changes as much as i can. Finally, I am happy to address any further issue you see it need to be modified. 

Hope to hear from you seen 

Thank you 

File attached Responses 

Introduction

1. Well written

2. Please emphasize the novelty of the study and provide two or more related works in the context of Palestine. 

Studies in context of Palestine were already used but not explained in terms of novelty and defects – so the following has been changed “Studies investigating Palestinian women's maternal health status and risk factors associated with adverse pregnancy outcomes are scarce which highlighted the great need to conduct such kind of research. These studies and reports have stated that Palestinian population has high prevalence of several pregnancy complications including gestational diabetes and hypertension but have never stated the exact percentages or addressed the living and socioeconomic circumstances as risk factors (15)(16, 19).” 

In addition, there is an obvious lack of research on the occurrence of various hypertensive disorders of pregnancy (HDP) in Palestinian women; they are not even included in the Palestinian health annual reports published by the Palestinian ministry of health (MOH) (40)

Methods

1. Data exclusion criteria explained well.

2. Please provide a detailed explanation for why we did not consider different available methods for imputing missing values in electronic health data or health data records, given that missing data in your final samples is a significant concern. 

Adding these values will make no difference to the findings of the research as most of the excluded pregnant women had only an initial booking appointment at which some had measured their blood pressure only and minority has done blood test & urine tests then they never come back. Pregnant women should follow up monthly with the antenatal clinics; however, some of them have attended their booking appointments only and never came back. We have tried to include every beneficial pregnant to the sample to make it as general as possible so we have included any pregnant women who at least attended the booking and a late trimesters appointments. Even though the number of excluded pregnant women who has lost to follow-up seems significant but it was not possible to avoid it. These cases had either only an initial visit or no third trimester visit which will not add any value to the results as pregnancy complication are diagnosed after 20 weeks of gestation.

3. Extend Data analysis a bit more, with more specification.

I can confirm to the editor that we have considered multiple analysis approaches and methods. Initially the data was analyzed without excluding the lost to follow up data, and then it was obvious that they had no beneficial values to be kept. Also several descriptive and analytical methods were used (regression, correlations, ... mixed models) to extend the research results. After all, we presented the best significant results in the forms of 6 tables. We can provide the editor with whatever analysis idea that he would like to check and we will be happy to do whatever necessary. 

Results- Well written

Discussion –

1. Well written.

2. Please make it a bit short in length and focus on your objective and future implication in terms of Palestine.

Kindly see the discussion after multiple sentences have been removed and other addressing objectives and future implications were added

Conclusion – well written

------------------ Letter Comments 

• There is a lack detailed methodological information in the abstract Thank you for bringing this to our attention It has been detailed

Methods: Medical records (N=7889) for pregnant women at primary healthcare centers in the North West Bank were reviewed for sociodemographic and medical data along with the reported fasting blood sugar, hemoglobin, and blood pressure in the first and second trimesters from July 2018 to July 2020. However, only 6640 were included in the analysis as 1249 were excluded for having multiple pregnancy or lost to follow up. Complications and risk factors were defined according to the available global guidelines. Then, descriptive analysis was used to show the percentages of different risk factors and complications among them. The correlation between the several characteristics and variables with these complications was assessed by calculating the odds ratios (OR) using logistic regression. P-values of <0.05 were considered significant. 

• More background information of the dataset could be added. 

it has been updated.

• Present the key variables and how they are measured in a tabular format

I had done this according to your request you can check figure 1.

• In adjusted model, kindly mention the variables were adjusted

Done i have changed it all over the tables, foot nails, and the result’s text.

• Add some strengths after the limitations. Done the following

However, this study included a prominent number of pregnant Palestinian women, which is unprecedented in previous studies. It is also the first one to study Palestinian society from all available aspects, including laboratory, clinical, obstetrical, and medical data along with social and living conditions. Then, we have linked these characteristics to the complications that it carries. Therefore, It will be the core for further studies monitoring Palestinian women's health and as an alert to give more attention to some complications in specific groups of the population. Also, actions should be taken urgently to reduce the high prevalence of these complications or at least address it early. Finally, this study is also a realistic example of the possibility of using the WHO guidelines in the Palestinian community.

Reviewer #1: 

Epidemiology is not clearly defined. Risk factors are not properly stated. We tried to make it as clear as possible now by adding more sentences to the methodology and add tabular figure for variables.

Grammatic errors (Some sentence in first paragraph is in future tense) and mistake in certain areas regarding the year of study. Mistakes in Table headings. Results need more specific explanation on associated factors and risk factors. The year’s mistake has been found and fixed, grammars mistakes were tried to avoided as much as possible but English is our second language. Even though we use grammar checker service during writing it is still not very accurate, we have tried to fix language issues as much as we can and we are happy to fix any if you could just highlight it for us. 

------------------- Reviewer #2 

All his comments were addressed above in the first page of this file

------------------ Reviewer #3: 

Abstract: Topic sentence(1st sentence) should rewrite for impressive/catchy

The results will be compared between women residing in urban or rural areas and those in refugee camps- Should change as the study has been already completed, however mentioning urban or rural -which should be clearly defined . Sentence has been changed to more attractive one, The future tenses has been changed to past tenses. urban and rural were exchanged until they are defined in the methodlogy

Introduction: dissimilar font size- please check carefully whole manuscript.

This issue has been addressed all over the manuscript

Methodology: The approach and procedure for gathering data for a retrospective medical study are explained in this section. The study has a few weaknesses in addition to its strengths as follows

1.Data Collection Period: From July 2018 to July 2020, research data were gathered. Although this is a significant amount of time, a longer duration of data collecting would have strengthened the results as a retrospective study This would give a more thorough grasp of the patterns and alterations across time.

The duration was for 2 years and it took researchs about one year to collect data, thank you for addressing this

2. The study was carried out during the pandemic covid -19, which had a major effect on patient accessibility and healthcare services. How this might have impacted the data and whether any modifications were taken to take these particular conditions into account are not addressed in detail in the study. 

The main current object of the study is to address the current situation of Palestinian pregnant, it will be great idea for us to run a future research to compare the effect of covid ( as a before / during / after ) but currently we couldn’t as covid spread in different timely patterns and waves and this will get us away out from the scope of the study. However we have highlighted this in methods and discussion , for example :-

The COVID-19 pandemic has worsened the circumstances for pregnant women, especially those residing in rural areas. Half of the study time coincided with the pandemic, further limiting access to healthcare for women attending public sector services, and to a lesser extent, for those who could afford private sector services

3. The study's exclusion criteria, which include removing expectant mothers who are carrying multiples, have undergone an abortion, or are lost to follow-up, may result in selection bias. The selection criteria for this exclusion may not have been well thought through, and it's probable that pregnant women in general are not adequately represented by it.

Thank you for caring about this point, as i mentioned for the editor we have try our best to include every possible pregnant women, but we were clear and honest about our results. Multiple pregnancy is considered as high risk pregnancy and it follow up with gyn-obs specialist with more intesive care and monitioring as they carries high risk for complication. So it was better to exculde and their size was significantly high in regard to sample size. The major portion was due to abortion or lost to follow up – in this section we can’t do any thing it is obligated for us to exclude these. for example a pregnant who has only did the booking or had only one BP measurment ! ( we need at least 2 ). So we tried our best to make the sample general as possible despite the hard circumstances

4.The study notes that there was a significant decrease in the number of beneficiaries receiving antenatal care in 2020. This decrease could be a crucial factor affecting the study's results, but the passage does not elaborate on how this change was addressed in the analysis.- Temporal changes.

We thought that the decrease is could be related to the Covid Pandemic and we have wrriten a paragraph to address this issue as short as we can as it is out of the scope of the study which is to determine the percentages of multiple risk factors and complications. In addition, nothing for sure can explain why did this happen? it could be due to covid, palestine isreali conflict, financial circumstance .. so this issue could be addressed by further research. However, i will be happy to add and address any specific point you have 

The limitations of the study should mentioned.

It is mentioned but it was in the middle of discussion , now it is in the bottom.

---

## [Decision Letter · Decision Letter 1]

12 Mar 2024

The epidemiology of gestational diabetes, gestation hypertension, and anemia in North Palestine from 2018 to 2020: A retrospective study.

PONE-D-23-29769R1

Dear Dr. Abumohsen,

We’re pleased to inform you that your manuscript has been judged scientifically suitable for publication and will be formally accepted for publication once it meets all outstanding technical requirements.

Kind regards,

Ibrahim Sebutu Bello, MBBS, MPH, MD, FMCGP

Academic Editor

PLOS ONE

Additional Editor Comments (optional):

All issues raised by reviewers have been addressed

Reviewers' comments:

Reviewer's Responses to Questions

**Comments to the Author**

1. If the authors have adequately addressed your comments raised in a previous round of review and you feel that this manuscript is now acceptable for publication, you may indicate that here to bypass the “Comments to the Author” section, enter your conflict of interest statement in the “Confidential to Editor” section, and submit your "Accept" recommendation.

Reviewer #1: All comments have been addressed

Reviewer #2: All comments have been addressed

Reviewer #3: All comments have been addressed

2. Is the manuscript technically sound, and do the data support the conclusions?

Reviewer #1: Partly

Reviewer #2: Yes

Reviewer #3: Yes

3. Has the statistical analysis been performed appropriately and rigorously? 

Reviewer #1: Yes

Reviewer #2: Yes

Reviewer #3: I Don't Know

4. Have the authors made all data underlying the findings in their manuscript fully available?

Reviewer #1: Yes

Reviewer #2: Yes

Reviewer #3: Yes

5. Is the manuscript presented in an intelligible fashion and written in standard English?

Reviewer #1: Yes

Reviewer #2: Yes

Reviewer #3: No

6. Review Comments to the Author

Reviewer #1: Better to make change in Title which will point to your methodology also.The present title lacks clarity.

Reviewer #2: This study will contribute significantly to understanding Palestinian women's maternal health. All comments have been addressed clearly by the authors.

Reviewer #3: Based on my the query, the authors addressed well , but there's room for improvement in terms of font style and size. Additionally, there's a need to minimize grammatical errors. Given that the study has already been completed, it's essential to ensure consistency in the tenses used in the abstract. There's also a need for overall enhancement in the write-up. As for example, "studies showed that anemia and high hemoglobin (Hb) levels during pregnancy affect the quality of the mother's delivery and her baby's health." Starting with such generic phrases can weaken the impact of a sentence.

7. PLOS authors have the option to publish the peer review history of their article (what does this mean?). If published, this will include your full peer review and any attached files.

Reviewer #1: **Yes: **Sreeja Savithriamma Aravindakshan

Reviewer #2: No

Reviewer #3: No

---

## [Editor Report · Acceptance letter]

20 Mar 2024

PONE-D-23-29769R1 

PLOS ONE

Dear Dr. Abumohsen, 

I'm pleased to inform you that your manuscript has been deemed suitable for publication in PLOS ONE. Congratulations! Your manuscript is now being handed over to our production team.

Kind regards, 

on behalf of

Dr. Ibrahim Sebutu Bello 

Academic Editor

PLOS ONE